# Intelligent, Flexible Artificial Throats with Sound Emitting, Detecting, and Recognizing Abilities

**DOI:** 10.3390/s24051493

**Published:** 2024-02-25

**Authors:** Junxin Fu, Zhikang Deng, Chang Liu, Chuting Liu, Jinan Luo, Jingzhi Wu, Shiqi Peng, Lei Song, Xinyi Li, Minli Peng, Houfang Liu, Jianhua Zhou, Yancong Qiao

**Affiliations:** 1School of Biomedical Engineering, Shenzhen Campus of Sun Yat-sen University, No. 66, Gongchang Road, Guangming District, Shenzhen 518107, China; 2Key Laboratory of Sensing Technology and Biomedical Instruments of Guangdong Province, School of Biomedical Engineering, Sun Yat-sen University, Guangzhou 510275, China; 3School of Integrated Circuits and Beijing National Research Center for Information Science and Technology (BNRist), Tsinghua University, Beijing 100084, China

**Keywords:** artificial throat, sound sensor, thermoacoustic effect, machine learning

## Abstract

In recent years, there has been a notable rise in the number of patients afflicted with laryngeal diseases, including cancer, trauma, and other ailments leading to voice loss. Currently, the market is witnessing a pressing demand for medical and healthcare products designed to assist individuals with voice defects, prompting the invention of the artificial throat (AT). This user-friendly device eliminates the need for complex procedures like phonation reconstruction surgery. Therefore, in this review, we will initially give a careful introduction to the intelligent AT, which can act not only as a sound sensor but also as a thin-film sound emitter. Then, the sensing principle to detect sound will be discussed carefully, including capacitive, piezoelectric, electromagnetic, and piezoresistive components employed in the realm of sound sensing. Following this, the development of thermoacoustic theory and different materials made of sound emitters will also be analyzed. After that, various algorithms utilized by the intelligent AT for speech pattern recognition will be reviewed, including some classical algorithms and neural network algorithms. Finally, the outlook, challenge, and conclusion of the intelligent AT will be stated. The intelligent AT presents clear advantages for patients with voice impairments, demonstrating significant social values.

## 1. Introduction

Verbal communication is the basic communication method of human beings. However, some patients in the world have deficiencies in language ability. In China, oral and oropharyngeal cancer accounts for approximately 307,000 new cases each year, constituting over half of the 572,000 new cases identified worldwide [1,2,3]. In addition to laryngeal cancer, diseases such as esophageal cancer and other unexpected accidents will also seriously affect patients’ linguistic ability. The interpersonal communication and quality of life of the mute people will also be seriously spoiled, leading to a negative impact on their mental and physical health [4]. Therefore, how to effectively reconstruct the vocal function of speech-impaired people, aiming at minimizing the detrimental effects of speech impairment, has become the focus of the whole society.

Nowadays, one of the most widely used pronunciation reconstruction methods is still the conventional electrolarynx [5]. The conventional electrolarynx (Figure 1a) [6], composed of motorized transducers with large rigidity, volume, and complex structure, can assist mute people to emit sound [7]. The working principle of the conventional electrolarynx is initially creating vibrations of the oral or pharyngeal at a constant fundamental frequency. Then, these vibrations will be transmitted to the throat or mouth. Following this, after interacting with the vocal cord tissue, the patient can produce audible speech [5,8]. The usage is shown in Figure 1b. Currently, most of the currently available electrolarynx has been adapted, designed, and modified on this working principle. For example, Isshiki et al. proposed an electrolarynx with better performance in voice [9]. Wu et al. put forward a method of solving the problem to eliminate the abnormal acoustic properties [10].

However, despite continuous changes and improvements, the conventional electrolarynx still has many shortcomings that have not been effectively addressed. First, the conventional electrolarynx is a hand-held device that takes up a person’s hand and restricts the normal movement of the hand. Second, managing the conventional electrolarynx is challenging. Initially, the patients often need to spend much time finding the most suitable site to attach the electrolarynx to the neck muscles. In addition to the proper site, the interface between the conventional electrolarynx and the skin significantly influences its vocalization. Consequently, achieving the appropriate tightness to adhere the electrolarynx to the neck muscle is also a time-consuming process for patients. Third, the conventional electrolarynx is only capable of producing a mechanized and monotonous sound, which lacks variation compared to natural sounds. This poor quality of sound may seriously affect the patient’s voice expression and communication experience, limiting their ability to communicate fluently. Therefore, the conventional electrolarynx has significant deficiencies in sound quality and learning difficulty, which affects speech recovery in patients with laryngeal cancer or laryngectomy.

In recent years, with the continuous development of materials science, solid-state physics, and electronic engineering, new methods have been provided to solve the difficult problems plaguing the development of the conventional electrolarynx [11]. In conventional electrolarynx, it is limited to being used as a sound emitter. However, the latest developing artificial throat (AT) can not only be used as a sound emitter but also as an intelligent sound sensor, integrating sound perception technology with the assistance of algorithms. As for the sound sensor, capacitive, piezoelectric, electromagnetic, and piezoresistive materials have been widely leveraged in the field. Some sound sensors are self-powered and can be combined with other physiological signs. In terms of sound emitters, conventional sound emitters typically rely on the electromagnetic effect with rigid and solid materials. In contrast, thermoacoustic materials are often flexible and thin, which could be fabricated as thin-film sound emitters. This characteristic makes them exceptionally well-suited for wearable applications. In contrast to the conventional electrolarynx, the latest AT developed based on new materials is lightweight, thin in thickness, simple in structure, soft in use, and comfortable to wear. These attributes collectively contribute to an enhanced patient experience.

As mentioned before, with the assistance of a machine learning algorithm, the AT has progressively gained intelligence in speech detection and recognition, addressing the deficiency of the electrolarynx in this regard. In 2020, Jin et al. developed a model with a large amount of data to recognize the long vowels and short vowels of human pronunciations, and the recognition accuracy can reach 83.6% in the long vowels, and 88.9% in the short vowels. Afterward, many other models have also been used in the speech recognition of AT, such as SR-CNN [12], AlexNet [13,14], Inception V3 [13], SCNN [15], etc. Obviously, machine learning algorithms have greatly assisted AT in breaking through the deficiencies of the electrolarynx in receiving voices. Moreover, AT combined with a machine learning algorithm brings these speech-impaired patients more effective assistance than the conventional electrolarynx.

In this review, we aim to give an overview of the intelligent flexible AT, which consists of the sound emitter, sensor, and recognition algorithm (Figure 2). The concept and composition of the AT are initially elucidated, and compared with the conventional electrolarynx. The second part focuses on the sound sensor, which covers not only sound but also other physiological signals, including electromyographic (EMG) signals that reflect voice information. In the third section, a thin-film sound emitter based on the thermoacoustic effect will be discussed, which can help the user to emit sound. In the fourth section, a detailed review of the intelligent AT, supported by various algorithms for sound wave recognition, will be stated, such as some digital signal processing techniques, classical machine learning algorithms, deep learning algorithms, and so on. Finally, the outlooks and limitations will be given. This review is helpful for the researchers who intend to study the AT devices.

## 2. Sound Sensor

When a sound emitter vibrates, such as the vocal cords of a person, the strings of a musical instrument, or other objects, it induces the surrounding medium to generate alternating zones of compression and rarefaction. This phenomenon gives rise to the production of sound. Sound can be regarded as a combination of simple harmonic waves, which can propagate through a solid, liquid, or gas [21,22,23,24]. When a sound wave strikes the human’s ear membrane, the different frequencies of sound result in various levels of vibration; only the frequencies ranging from 20 Hz to 20 kHz can stimulate the human nervous response to produce the hearing sensation [25,26]. Within this process, the human ear plays a key role, acting as a sound sensor with a sophisticated structure. Owing to the sophisticated design, human ears have a high sensitivity to voice. In the realm of sound sensors, the material and structure also need to be specially designed to detect sound, especially weak sound.

Recent decades have witnessed the rapid development of sound sensors. Capacitive, piezoelectric, electromagnetic, and piezoresistive sensors have been widely studied. Moreover, the design of sensing materials and the preparation of sensors have significantly improved. Furthermore, when integrated with other human physiological signals, sound detection has achieved high levels of accuracy.

### 2.1. Capacitive Sound Sensor

The capacitance for the common plain plates can be expressed as
(1)$$C=\varepsilon \frac{S}{d},$$
where ε is the dielectric coefficient of the medium between the plates, depending on the physical property of the material between the plates. S stands for the area of one plate, and d is the distance between the plates. Derived from this formula, the capacitive sensor can be divided into three categories: the first category is based on a variable dielectric coefficient, when the dielectric changes, the capacitance will change. Many humidity sensors are developed on this basis. The second category considers the changes in area [27,28,29]. The third category is building on the shift in distance. Considering the characteristics of the sound waves, many capacitive sensors are based on the distance changes between movable plates and other fixed plates [30].

The principle of the third category of capacitive sound sensors can be stated as follows: A bias voltage is initially applied to load the plates. Subsequently, the voltage between the two plates remains constant unless an incoming sound wave induces vibrations in the movable plates. This vibration leads to a change in capacitance and, consequently, a variation in voltage [31]. In this manner, the capacitive sound sensor senses the acoustic signal, transforming it into an electrical signal with a flat frequency response [32].

Lee et al. realized a sound sensor with a sophisticated capacitive structure [33]. When the device is attached to the neck skin, the vibration of the neck muscles will cause changes in the capacitance between the movable plates and the fixed plates (Figure 3a). When connected to the capacitance sensing circuit that effectively converts capacitance changes to voltage variations, the vibration of the neck muscles will be transformed into an electrical signal (Figure 3b). Compared with the conventional microphone, the device could effectively resist noise interference, owing to it recognizing the voice just by the skin vibration (Figure 3c).

### 2.2. Piezoelectric Sound Sensor

The ability to generate an electrical charge by applying mechanical stress is called the piezoelectric effect [34], which can be found in some materials, such as polyvinylidene difluoride (PVDF) [17,35,36,37], lead zirconium titanate (PZT) [36,38,39,40], zinc oxide (ZnO) [41,42,43]. Compared with capacitive sensors, piezoelectric sensors do not require an additional bias voltage to be supplied and do not require additional circuit design. Furthermore, a quantity of sensors can also be self-powered.

Lang et al. developed a PVDF-based sound sensor using electrospinning technology (Figure 3d) [17]. Figure 4a schematically illustrates a proposed sound-sensing mechanism. When the sound wave hits the sensor, the sound absorption induces vibration of the nanofiber network, the Au layer, and the polyethylene terephthalate (PET) sheet. Part of the nanofiber mesh is covered by a PET sheet and Au layer, but part of the nanofiber mesh is directly exposed to the sound absorption. The directly exposed part vibrates more intensively than those covered, causing asymmetric vibrations on the propagation along the fiber and a heightened sensitivity in sound perception. In addition, the piezoelectric sound sensor has good sound perception at low frequencies. Figure 4b-ii demonstrates its ability to effectively distinguish between two different low frequencies, approximately 190 Hz and 260 Hz. However, its measured sound pressure after 400 Hz will rapidly drop to 0, so its performance at high frequencies is notably inferior to that of the capacitive sensor [32,37]. Recent studies have shown another key physical property of this material: the thickness of the piezoelectric material plays a key role in sound detection. Lim et al. fabricated a piezoelectric sound sensor with single-walled carbon nanotubes (SWCNTs) and a PVDF network with varying thicknesses. They discovered that the output voltage and impedance would show a nearly linear relationship within a proper thickness. If the thickness is too small (below 200 μm), the impedance will drop rapidly, possibly linked to a short circuit inside the electrostatic spinning filament [37].

Owing to the fact that in piezoelectric sensors, the mechanical input can be converted directly into an electrical output with no external power source required, this type of sensor is considered to be self-powered, which is promising in fields like sound energy harvesting [44,45]. The working process of the self-powered device can be explained in Figure 5a [32,46]. Figure 5a-i schematically illustrates the structure of the spring-substrate nanogenerator, which is composed of a metal spring. The metal spring is composed of an Ag electrode and a quantity of ZnO nanowires which are passivated with polymethyl methacrylate (PMMA). When a tiny plate weighing 15.2 N is placed on the self-powered sensor, the piezoelectric sensor will produce an output voltage of 0.23 V. Furthermore, Figure 5a-iii shows an almost linear relationship between the output voltage and current, exhibiting a sensitivity of 2.8 nA × kg^−1^ and 45 mV × kg^−1^. This remarkable sensitivity indicates the sensor’s performance which has been employed in piezoelectric sound detection.

Cui et al. achieved a sound-driven triboelectric nanogenerator (TENG) based on the piezoelectric material PVDF (Figure 5b,c). Their device demonstrates the capability to instantaneously illuminate 138 LEDs, as shown in the inset of Figure 5d, in response to a 114 dB/160 Hz sound [47]. The energy generated by this nanogenerator can not only light the basic electronic components, like LEDs, but also power other commercial products. This indicates that these sound sensors have an exceptionally wide range of applications. Shao et al. fabricated a single-layer piezoelectric nanofiber sound sensor utilizing PET films, Au electrodes, and PAN-PVDF fiber membrane (Figure 5e). This sensor can power the calculator to perform the calculation process and charge the capacitor (Figure 5f) [48].

### 2.3. Electromagnetic Sound Sensor

The classic structure of electromagnetic sensors is often composed of a coil, a diaphragm, and a permanent magnet. When the diaphragm is vibrated by sound waves, the diaphragm will drive the coil to move in the magnetic field, thus generating output current. However, this classic structure is mostly rigid and only partially flexible due to limitations such as coils and magnets [49,50,51].

As a result of advancements in manufacturing methods for achieving flexible magnetic membranes, Huang et al. successfully produced flexible neodymium magnet (NdFeB) membranes in the origami approaches in 2019 [52]. In 2020, they manufactured an additional fully flexible electromagnetic sensor incorporating NdFeB (Figure 6a), enabling repeated bending and twisting for attachment to the body [18]. Moreover, this fully flexible structure can serve as a sound sensor by utilizing the electromagnetic induction between the copper coil and magnetic membrane to detect the vibration of the vocal cords (Figure 6b,c).

### 2.4. Piezoresistive Sound Sensor

The piezoresistive effect refers to the change in resistance of a material of semiconductor or metal when mechanical strain is applied [53,54,55,56]. The piezoresistive effect in metals and semiconductors was discovered in the 18th century and 19th respectively. For some electrical conductors with the same physical property when measured in different directions, the relative resistance change can be derived as
(2)$$\frac{\Delta R}{R}=\frac{\Delta l}{l}\left(1+2v\right)+\frac{\Delta \rho }{\rho },$$
where l describes the length of the electrical conductor, v is Possion’s ration of the electrical conductor, and ρ is the resistivity. As described in this formula, the change of the length $\left(\Delta l\right)$ and the variance of the ρ $\left(\Delta \rho \right)$ determine the change of resistance $\left(\Delta R\right)$ [54,57].

As depicted in Figure 7a, the application of an external force induces compressive deformation in the sensor within increasing material contact, which creates additional conductive paths and varying resistance. In recent years, studies have indicated that strain sensors fabricated with nanomaterials, when exposed to sound waves, generally exhibit the piezoresistive effect and are widely recognized as sound sensors. These nanomaterials include graphene [55,58,59], carbon nanotube (CNT) [60,61,62], MXene [19,63,64,65,66], and so on. Tao et al. fabricated a sound sensor with graphene, derived from the graphene resistance changes in response to applied forces [58]. Ma et al. developed a piezoresistive sensor capitalizing on highly ordered hierarchical architectures of hybrid 3D MXene/reduced graphene oxide (MXene/rGO) (Figure 7b). This design combines the large specific surface area of graphene oxide with the excellent conductivity of MXene, enabling the sensor to recognize a wider range of pressures and detect the vibration when attached to the throat (Figure 7c) [19]. Gong et al. proposed an ultrathin gold nanowires (AuNWs) impregnated tissue paper sandwiched between a blank PDMS sheet and a patterned PDMS sheet (Figure 7d), which could achieve the sensitivity of 1.14 kPa^−1^. As illustrated in Figure 7e,f, when a voltage is fixed, the application of external pressure results in a decrease in resistance and an increase in current [67].

### 2.5. Silent Speech Interfaces in Sound Recognition

Alternative methods for sound recognition exist for scenarios where sound signals are unavailable. One such method is the silent speech interface, a system capable of generating a digital representation of speech by acquiring sensor data during the human speech production process [68]. For instance, some physiological signals during speaking are relevant to vocalization information, which can serve as a sensor in the silent speech interface [69,70,71,72]. Tanja Schultz et al. obtained a high recognition rate of oral word signals by using EMG signals [73]. Liu et al. fabricated a tattoo-like patch to acquire the EMG from three muscle channels to recognize the instructions [74]. Compared with the acquisition of only a single resistance or voltage signal, the simultaneous acquisition of the EMG signal and other physiological signals will bring great help to the subsequent signal processing and speech recognition [75,76]. Tian et al. proposed a dual-channel speech recognition system based on the EMG and mechanical sensors. In comparison to utilizing the mechanical signal of the neck muscles, the signals including the movement of the neck muscles and EMG exhibit superior performance in sound detection and recognition [77]. In addition to EMG, electroencephalographic (EEG) also plays a role in speech recognition. Pradeep Kumar et al. developed a speech recognition framework with the help of EEG signals with high accuracy in the recognition task of 30 text and not-text classes [78]. Anne Porbadnigk et al. also investigated the use of EMG by utilizing 16 EEG channels with a 128-cap montage for speech recognition [79]. Apart from the physiological signals, silent speech interfaces also include some real-time characterization of the vocal tract methods such as using ultrasound and optical imaging of the tongue and lips for speech recognition.

As stated before, the sensing principles for the latest generation of flexible and wearable sound sensors including capacitive, piezoresistive, piezoelectric, etc. The present sensors have been gradually miniaturized and flexible with soft, highly curved properties, playing a crucial role in the voice recognition capabilities of AT. However, the current AT still have problems such as sensitivity and accuracy, which need to be further improved. Additionally, optimal performance and durability during use also need to be focused [57].

## 3. Sound Emitter

The sound emitter is an important component of the AT. The successful restoration of patients’ voices with the help of AT relies on the performance of the sound emitter [80]. A sound emitter is a transducer that converts electrical signals into sound signals. Taking the most common moving-coil sound emitter as an example, audio-electrical signals are transmitted through an electromagnetic effect. This effect induces vibrations in its diaphragm, resonating with the surrounding air and generating sound. However, the electromagnetic effect requires a permanent magnet, a coil, and a diaphragm to create vibrations in the air and then produce sound. As a result, moving-coil speakers are typically larger in size [6]. In addition to utilizing the traditional vibrating sound emitter, the AT can also produce sound by means of the thermoacoustic (TA) effect [81]. This sound emitter is only a thin film with a small size. Due to this principle, the AT can be worn directly on the patient’s larynx as an electronic skin [82].

The sound emitters based on the TA effect is a device that generates sound using heat. The physical process of the TA effect can be described as follows: when an alternating current signal passes through a thin metal film, the film generates Joule heat, which is rapidly transferred to the surrounding air medium. Due to the periodic rise and fall of the temperature of the surface of the metal, the air molecules in the thin layer of the surface of the metal are constantly expanding and contracting, thus generating sound waves. By controlling the rate of heating and cooling, the frequency of the sound produced can be modulated, allowing for the generation of different sound intensities and tones [11,83].

### 3.1. Development of the TA Sound Emitter

The TA effect was discovered more than 200 years ago. In the 18th century, Byron Higgins experimented with a hydrogen flame placed in the proper position in a vertical tube with openings at both ends, and sound was produced in the tube. This is historically known as a singing flame and was the first discovery of the thermoacoustic effect.

In 1917, Arnold and Crandall proposed a TA sound emitter made of suspended 700-nm platinum film (Figure 8a) [84]. Then, they analyzed its sound-emitting mechanisms theoretically. When an AC current with the sound frequency passes through the surface of a platinum film with a low heat capacity, the heat is transferred to the ambient air, causing the air to expand periodically, thus producing sound. In their theory, the formula of the SP can be derived as
(3)$$P_{rms}=\frac{\sqrt{\alpha }\rho_{0}}{2\sqrt{\pi }T_{0}}\times \frac{1}{r}{\times P}_{\mathrm{input}}\frac{\sqrt{f}}{C_{s}},$$
where $C_{s}$ is the heat capacity per unit area (HCPUA) of the thermoacoustic thin film, and *f* is the frequency of the excitation frequency. $P_{\mathrm{input}}$ and *r* are the input power and the distance between the thin film with the microphone, respectively. α, $\rho_{0}$, and $T_{0}$ are the thermal diffusivity, density, and temperature of the ambient gas. This equation indicates that the sound pressure produced by the TA sound emitters increases with smaller HCPUA, higher frequency, and input power.

Their theoretical model led to the derivation of the basic sound generation equation. However, over the subsequent 100 years, the TA sound emitter has been overlooked, due to the specific properties of materials and the limitation of signals being only at 4 kHz, coupled with low sound pressure. Nonetheless, it is crucial to acknowledge that this theoretical groundwork provides a foundation for the subsequent development of the thermal.

It was not until 1999 that H. Shinoda et al. extended Amold et al.’s surface-sounding theory. They introduced a porous-silicon-based sound emitter in Nature (Figure 8b) [81]. This approach involved applying a 30-nm-thick aluminum film on top of a 10-µm-thick, porous silicon layer, resulting in a wide-band sound emitter capable of achieving a notable sound pressure of 0.1 Pa (1–100 kHz). In this work, they enhanced and refined the previous module. In their theory, the SP can be described as
(4)$$P\left(x,\omega \right)=\sqrt{\frac{\gamma \alpha_{a}}{C_{a}}}\frac{P_{A}}{vT_{A}}\times \frac{\exp\left(-jkx\right)}{\sqrt{\alpha C}}\times q\left(\omega \right),$$
where $P_{A}$ is atmospheric pressure, $T_{A}$ is room temperature, v is the sound velocity, γ = $\frac{C_{p}}{C_{v}}$ = 1.4, $C_{p}$ is the heat capacity at constant pressure, $C_{v}$ is the heat capacity at constant volume, k is the wavenumber of sound in free space. $\alpha_{a}$ is the thermal conductivity in air, and $C_{a}$ is the HCPUA.

In the 21st century, the development of nanotechnology has led to breakthroughs in *T_A_* sound emitter devices. In 2008, Xiao et al. achieved a groundbreaking in thermoacoustic theory [83,85]. They fabricated a sound emitter utilizing CNT (Figure 8c). This device boasts a wide frequency response range and a high SPL, due to the low HCPUA of CNT. However, their experimental results did not align with Arnold and Crandall’s theory, then they identified that Arnold’s theory neglected the rate of heat loss per unit area of the thin film and the instantaneous heat exchange per unit area. Based on this observation, they proposed their own model as follows:(5)$$P_{\mathrm{rms}}=\frac{\sqrt{\alpha }\rho_{0}}{2\sqrt{\pi }T_{0}}\times \frac{1}{r}{\times P}_{\mathrm{input}}\times \frac{\sqrt{f}}{C_{s}}\times \frac{\frac{f}{f_{2}}}{\sqrt{{\left(1+\sqrt{\frac{f}{f_{1}}}\right)}^{2}+{\left(\frac{f}{f_{2}}+\sqrt{\frac{f}{f_{1}}}\right)}^{2}}},$$

In their new module, two constants $f_{1}$ and $f_{2}$ were added, $f_{1}=\frac{\alpha \beta^{2}}{\pi k^{2}}$ and $f_{2}=\frac{\beta_{0}}{\pi C_{s}}$. Additionally, the previous Arnold’s theory is only suitable for higher HCPUA and is not applicable to smaller HCPUA. Xiao et al. introduced a modified model that overcame these limitations. Based on their theory’s findings, they fabricated a CNT thin film TA sound emitter, which possesses the merits of nanometer thickness and are transparent, flexible, and stretchable [83].

In 2010, Hu et al. modeled a TA sound emitter in the low and high-frequency bands on the basis of H. Shinoda [86], confirming that there exists a very wide range of constant amplitude-frequency response mostly in the ultrasonic region for TA emission from any solid. In the same year, V. Vesterinen et al. verified the theoretical model by using nanoscale aluminum as a sound-emitting layer (Figure 9a) [87]. Then, they concluded that the primary factor influencing sound pressure in the low-frequency band is the properties of the substrate, whereas in the high-frequency band, the material’s heat capacity is the predominant major. However, there are still some defects in Hu’s model. In 2011, Tian et al. prepared graphene as a thermoacoustic device by means of chemical vapor deposition (CVD) (Figure 9b) [88]. Then, they elucidated the relationship between the surface temperature of the sound-emitting layer and the applied energy. Their experimental results are not in line with the previous module. They found that in Hu’s module, they omitted the 30 nm aluminum which functioned as the heat source. They assumed that the conductor was thin enough for this aspect and could be neglected. Owing to these findings, they modified their module as follows:

For $f<\frac{a_{s}}{4\pi L_{S}^{2}}$ at low frequencies in far-field, the SP could be derived as
(6)$$P_{rms}=\frac{R_{0}}{\sqrt{2r_{0}}}\times \frac{\gamma -1}{v_{g}}\times \frac{e_{g}}{M\left(e_{s}+a_{c}\right)+e_{g}}\times q_{0},$$
for $f>\frac{a_{s}}{4\pi L_{S}^{2}}$ at low frequencies in far-field, the SP could be derived as
(7)$$P_{rms}=\frac{R_{0}}{\sqrt{2r_{0}}}\times \frac{\gamma -1}{v_{g}}\times \frac{e_{g}}{\left(e_{s}+a_{c}\right)+e_{g}}\times q_{0},$$
where f is the frequency of voice; $\alpha_{s}$ and $L_{s}$ is the thermal diffusivity and thickness of the substrate, respectively; $r_{0}$ is the distance between the TA sound emitters and the microphone for the test; γ is the heat capacity ratio in gas; $\upsilon_{g}$ is the velocity of voice in gas; $e_{i}$ is the thermal effusivity of material which is determined by material; $q_{0}$ is the input power density; M is a frequency-related factor.

Xie et al. also proposed a new model [89], based on the energy conversation, which is easy to analyze and calcite. They module can be displayed as follows:(8)$$p_{rms}=\frac{m_{air}\cdot f\cdot \dot{\left(Q_{air}\right)}}{2\sqrt{2}C_{p}T_{0}r},$$
where $m_{air}$ is the molecular weight of air, f is the frequency of the acoustic, $Q_{air}$ is the thermal energy diffused into the air, $C_{p}$ is the heat capacity at constant pressure, $T_{0}$ is the room temperature and r is the measuring distance from the source. Tao et al. verified the correctness of the experiment (Figure 9c) [58].

### 3.2. TA Sound Emitter Made of Different Materials

A high-performance TA sound emitter needs to efficiently conduct heat into the air and convert it into sound. This imposes elevated requirements on the sound-generating material, which needs to have a very low specific HCPUA. In order to make high-performance TA sound emitter devices, three conditions should be satisfied. First, the conductor should be thin enough with a low HCPUA. Second, the conductor should be suspended to prevent thermal leakage from the substrate. Third, the conductor area should be large enough to build a sufficient sound field [90]. Various materials can be used in the construction of TA sound emitters, each with its own set of characteristics and applications. There are some common materials including graphene [90,91,92], MXene [11,93], CNT [94,95,96], metallic nanowires [97,98], and so on.

#### 3.2.1. Graphene

Graphene is an emerging two-dimensional material with high electromobility, high flexibility, and low heat capacity. It is very suitable to be applied in TA sound emitters. Graphene-based TA devices combine the advantages of graphene and TA sound emitter, exhibiting unique and excellent performance. The sonic frequency required will change as the frequency of the excitation voltage is altered.

CVD is a common technique for graphene preparation. In 2012, J. Suk et al. prepared a graphene film with excellent light transmission by CVD and fabricated it into a TA sound emitter (Figure 10a) [91]. Then, they demonstrated the effect of different substrates and areas of substrates on the sound pressure through experimental studies. For the first time, they improved the influence of the sound pressure from the membrane material to the flexible substrate, such as PET. Meanwhile, they experimented with the TA sound emitter with different curvatures, which opened a new application of the TA sound emitter in flexible devices. Using the same fabrication technique, CVD, Tian et al. prepared monolayer graphene, which has a defect-free structure and excellent light transmission and can be controlled in terms of the number of layers [90]. The monolayer graphene was then fabricated into graphene headphones (Figure 10b). Then they tested its delay, flatness, and power linearity. Due to its ultra-high frequency response, TA sound emitter headphones have been utilized in animal studies as signal transmitters, facilitating the future exploration of animal communication.

The frequency of graphene TA sound emitters is linked to the voltage and current applied. M.S. Heath et al. proposed a graphene-based ultrasonic TA sound emitter by combining various frequencies of alternating current applied to a thermoacoustic device to generate sound waves of different frequencies. The TA device was then made into a field effect tube, and the bias voltage was controlled to switch the TA sound emitter on and off and adjust the volume of the TA sound emitter [92].

The graphene sound emitter exhibits outstanding electromobility and flexibility, enabling its attachment to a person’s skin. In this capacity, it serves as a sound emitter in AT. In 2019, Wei et al. proposed a wearable skinlike ultrasensitive artificial graphene, which can serve as a sound emitter and can be directly attached to the larynx of the aphasic person (Figure 10c) [80]. In 2023, Yang et al. also fabricated an AT within a graphene sound emitter (Figure 10d) [13].

#### 3.2.2. MXene

MXene is 2D transition metal carbides or carbonitrides with the composition M_n+1_X_n_T_x_, where M is a transition metal; X is carbon or nitrogen; T represents surface functional groups such as -OH, =O, and -F; and n is an integer from one to four [85,102,103,104,105]. In particular, the abundant surface functional groups on MXene enable strong adhesion to various substrates. This capability allows the fabrication of mechanically stable flexible TA sound emitters, ensuring resistance to delamination from substrates during mechanical deformations [20].

In comparison to graphene, the MXene-based sound emitter device has a higher SP than that of graphene with the same thickness. Gou et al. fabricated MXene-based TA sound emitters using anodic aluminum oxide (AAO) and polyimide (PI) substrates (Figure 10e) [93]. These Ti_3_C_2_ MXene exhibits a higher SPL of 68.2 dB (f = 15 kHz) and displays a very stable sound output spectrum when the frequency varies from 100 Hz to 20 kHz.

The property of TA sound emitters based on MXene is stable. In a study conducted in 2023, Kim et al. successfully fabricated an ultrathin MXene-based TA sound emitter exhibiting consistent sound performance for 14 days (Figure 10f) [20]. Moreover, these sound emitters exhibit deformability in various configurations such as bent, twisted, cylindrical, and stretched-kirigami. They can be manipulated into diverse 2D and 3D shapes under different mechanical deformations.

#### 3.2.3. CNT

CNTs are cylindrical structures composed of carbon atoms with extraordinary electrical and mechanical properties. CNTs exist in various forms, including SWCNTs and multi-walled carbon nanotubes (MWCNTs), depending on the number of layers of carbon atoms [106]. CNTs have low HCPUA and high surface area per unit volume, which helps to generate high-level TA sound. In addition, the aerogel structure of CNT films facilitates the permeation of gas molecules, boosting its efficiency remarkably in sound emitting [16].

In 2015, Mason et al. observed the thermoacoustic transduction process at the single-molecule level, as illustrated in Figure 10g [99]. Leveraging this minimal length scale, they tested the assumptions made in previous models used to describe 2D thermoacoustic films. Additionally, they sought to establish correlations between the thermoacoustic efficiencies of these nanotube devices and their electrical impedance, aiming to gain insights into underlying loss mechanisms.

Similar to the previously discussed graphene TA sound emitters, when the HCPUA of CNT films is so low, the CNT TA sound emitters can also achieve very high SP. Romanov et al. fabricated TA sound emitters made of thin and freestanding films of randomly oriented SWCNTs (Figure 10h) [16] with a small HCPUA, the maximum frequency of the emitting sound can reach as high as 100 kHz.

#### 3.2.4. Metallic Nanowires

Metallic nanowires are extremely thin wires with diameters on the nanoscale, with many unique behaviors that have not been seen in bulk materials. [100,107,108,109,110]. Ag nanowires (AgNWs) are one kind of metallic nanowires, and there is high conductivity and transmittance in random networks. Utilizing this property, Tian et al. fabricated flexible, ultrathin, and transparent sound-emitting devices with a low driving voltage, as illustrated in Figure 10i [100]. However, the presence of nanowire–nanowire junctions within these devices poses challenges in precisely defining their lateral dimensions. In contrast to AgNWs, AuNWs exhibit distinct properties. They can be precisely defined lateral dimensions. Consequently, AuNWs allow for experimental performance comparison with theoretical predictions. By employing AuNWs, Dutta et al. prepared TA sound emitters consisting of arrays (Figure 10j) [101]. Their results fit with the classical theory proposed by Vesterinen et al. [87].

Due to the high intrinsic electrical conductivity of copper, copper nanowires (CuNWs) also represent a promising future in the TA sound emitter. Bobinger et al. fabricated TA sound emitters utilizing CuNWs [110], featuring an exceptional HCPUA of 1.9 × 10^−2^ J/(m^2^K), rendering them well-suitable for applications of TA sound emitters.

In summary, TA sound emitters have been invented and discussed since the early 20th century. Since then, these emitters have undergone significant evolution and refinement, paralleling the continuous advancements in material preparation techniques. In the process, materials have evolved from nanoscale aluminum layers to carbon nanotubes, and finally to graphene, which is now the dominant material. Furthermore, applications have also transitioned from the simplicity of basic TA sound emitters to their integration and expanded use, including sophisticated roles such as serving as sound emitters in intelligent AT.

## 4. Post-Processing and Recognition Algorithm

Based on sound detect devices mentioned in Section 2, vibration signals and other physiological signals can be collected directly. Semantic analysis of these signals is the ultimate purpose of AT, as these signals contain rich and crucial infseormation for communication [111,112,113,114]. The simplest way for recognition and distinction is directly observing the electrical output wave or capturing the wave with a microcontroller in the time domain [80,115,116]. Nevertheless, some throat vibration signals with similar pronunciations can be challenging to distinguish in the time domain. To accurately analyze semantic information, machine learning is an appropriate solution [72,117,118,119].

Depending on whether the input data is labeled, the machine learning algorithm can mainly be devised into supervised learning, where the input data is labeled, and unsupervised learning, where the input data is not labeled. The training set for the semantic recognition needs to be labeled, so most of the machine learning algorithm utilized is supervised learning algorithms, such as neutral network [120,121,122], support vector machine (SVM) [123], Bayes classification [124], and so on.

### 4.1. SVM

SVM is a supervised machine learning algorithm used for classification and regression tasks. SVM classifies data by constructing hyperplanes in a high-dimensional space. It represents samples as points, maximizing the gap between distinct categories. SVM adapts to complex patterns using kernel functions, making it suitable for diverse applications like image recognition and text classification. Moreover, due to its ability to identify decisive support vectors and eliminate numerous redundant training samples, SVM is a helpful tool for avoiding the “dimension disaster”. Fang et al. fabricated a PVDF flexible piezoelectric sensor to collect the throat vibration signals, utilizing the SVM to recognize and process the signals [114]. During the machine learning process, the number of training sets and test sets is very important which determines the accuracy and training cost. In this work, they discovered that when the number of training set samples and test set samples was 50 and 100, a very small sample, the training sets and test sets can represent the best performance. As for the hyper parameter, they adopted a heuristic method Grid Search-Support Vector Machine (GSSVM), which finds the appropriate hyper parameters through a grid search with a specified range and step size. As depicted in Figure 11a,b, the 3D viewer illustrates that when the penalty factor ‘c’ was set to 22.6274, the recognition accuracy reached its optimum level. The result of accuracy for speaker recognition and semantic recognition can reach as high as 95.97% and 97.5%, respectively.

### 4.2. Neural Network

Neural network is a computational model, inspired by the structure and function of the human brain, particularly the work principle of neurons. A convolutional neural network (CNN) is a specialized type of neural network that learns feature engineering by itself. The CNN with deep structures is adept at uncovering concealed intrinsic connections within the data and extracting abstract features effectively. The structure of the CNN consists of an input layer, convolutional layers, pooling layers, fully connected layers, and output layers. As a fundamental building block, the convolutional layers apply filters or kernels to extract the features in the input data. Moreover, the pooling layers follow the convolutional layers and are used to downsample the spatial dimensions of the input and reduce computation. In the end, the fully connected layers connect every neuron from the previous layers to the current layers and lead to the output layers that produce the final predictions. Jin et al. developed an MXene-based AT and harnessed CNN to accomplish the categorization task of distinguishing between long and short vowels (Figure 12) [12]. Owing to the deep structure of the neural network, a large amount of data is required. In this work, a total of 1500 data was adopted, including 750 long vowels and 750 short vowels. Among them, 1050 data were randomly selected as the training data set, and the others were used as the testing data set. After about 200 epochs of training, the result of accuracy for long vowels and short vowels reached 83.6% and 88.9% respectively.

### 4.3. Relief

Relief is a feature selection algorithm employed in machine learning and data mining. Especially beneficial when dealing with datasets containing numerous features, Relief aims to recognize the most crucial features for a predictive model. As an algorithm capable of identifying the most relevant features, Relief can collaborate with other feature extraction algorithms such as CNN and others, to identify valuable features, and reduce data dimensions. Yang et al. utilizing an integrated machine learning model proposed a graphene-based intelligent wearable AT for speech recognition and interaction [13]. In this work, they take advantage of the groundbreaking architectures within the realm of CNN, AlexNet for feature extraction, and introduce an improved AlexNet model. Furthermore, they choose Relief for feature selection and SVM as a classifier. As shown in Figure 13, the improved AlexNet extracts 10 features through the five convolution layers and other layers, then the Relief sorts the most important features for the SVM to classify. Compared with other models, including single AlexNet and another ensemble model (improved AlexNet + SVM), the model composed of improved AlexNet, Relief, and SVM attains significant enhancement in classification and time cost (Figure 14a). The result of accuracy can reach more than 90% in the task of recognizing daily words.

In addition to employing machine learning algorithms, mixed-modality is also harnessed in the signals acquisition and process, a mixed-modality signals can capture different aspects of information, resulting in enhanced accuracy and performance compared to using a single modality in isolation [76]. Qiao et al. applied the Au nanomesh as the physiological electrodes to detect EMG signal, while leveraging the Au/PU nanomesh as the strain sensor in the throat (Figure 14b). Furthermore, they introduced a synergetic CNN algorithm (Figure 14c) consisting of a modified CNN to analyze the EMG signals and a two-layer CNN to analyze the stress-strain, aiming at distinguishing voice signals. The result of the accuracy can reach as high as 98.9% (Figure 14d) [15].

In summary, the speech recognition function serves as the bridge connecting the sound sensor component of AT to the sound emitter component. Nowadays, the advancement of machine learning algorithms, including CNN, AlexNet, and other artificial algorithms, has significantly improved recognition accuracy and expanded the language corpus. This expansion has broadened the application landscape of AT.

## 5. AT Serving as a Sound Sensor and Emitter

This paper has introduced the three components of AT in early sections, namely, the sound-sensing part, the sound-emitting part, and the speech recognition part. However, it should be noted that the AT is not comprised of a single component; rather, it is a combination of the three parts.

Wei et al. developed a device that integrates both sound sensing and sound emission capabilities with speech recognition functions. They devised a system for sound sensing utilizing a custom-made circuit board (Figure 15a,b) and performed feature extraction in the time domain based on changes in resistance (Figure 15c), then they connected the AT to the microcontroller which transforms the changes of the resistances into different voltages. Variances in voltage will result in different sounds in the emitter section of the AT. Consequently, if the tester executes strong movements, there will be a significant voltage variation, causing the sound emitter to say “OK”. Conversely, if the tester’s movements are weak, the sound emitter will state “NO” [80].

Tao et al. utilizing the microcontroller also developed a device comprising a sound receiver and sound emitter with a voice recognition function in the time domain. Figure 16a shows the workflows of the recognition process. The microcontroller will initially detect the amplitude and the duration of the voice by capturing the resistance of the graphene AT until either the amplitude or time reaches the thresholds. Afterward, the digital function generator will be applied to the graphene AT for 3 s. In their device, different amplitude and last time will lead to the activation of different digital function generators, producing varied volumes and frequencies [58].

Qiao et al. further proposed AT combined with sound sensors, sound emitters, and speech recognition, as previously mentioned [15]. Utilizing machine learning algorithms for speech recognition, individual English letters, such as ‘B’, ‘C’, ‘D’, ‘E’, and ‘F’, can be discerned through the combination of EMG and strain sensor, Au/PVA nanomesh, in the neck muscle. Following the classification by the algorithms, the sound sensor is then repurposed as a sound emitter, producing the corresponding letter at an intensity of 78 dB.

With larger datasets and more sophisticated algorithms, as introduced in the paper previously, Yang et al. proposed an enhanced and more intelligent AT [13]. Their innovation extends the scope from identifying individual letters to recognizing complete sentences. Common everyday language sentences like ‘I’m back’, ‘I’m fine’, ‘What’s for breakfast’, ‘What’s for lunch’, and ‘What’s for dinner’ can be accurately identified, achieving a high correct rate in patients with a laryngectomy. Following the classification of these sentences by artificial intelligence algorithms, the AT can speak the corresponding sounds at an approximate intensity of 60 dB. Additionally, the device exhibits robust performance, effectively recognizing sentences in subtle sounds or noisy environments. Table 1 compares various devices with sound emitting and detecting functions.

In conclusion, AT is a combination of the sound sensor and sound emitter with a speech recognition function. Each part performs a specific function in speech recognition and sound emitting. In terms of sound sensors, a variety of materials, such as graphene and Au/Pu nanomesh, have been widely used in sound harvesting, which can effectively capture the physiological signals and vibration signals of the human body (Figure 16b) [15]. Then, artificial intelligence algorithms such as SVM, CNN, and Relief are applied to infer physiological signals from the voice signals, recognizing distinctive features. Following this, by utilizing thermoacoustic materials such as graphene film (Figure 16c) [88], the collected information will be output as audio signals. The AT commonly used is often a device that combines all of the three functions in one unit (Figure 16d–f) [58].

## 6. Challenge and Prospect

The recent research developments of intelligent flexible AT with sound emitting, detecting, and recognizing abilities are demonstrated above. However, many challenges still need to be overcome.

In the post-process and artificial algorithms, the current artificial algorithms used for AT are less general, and smaller in datasets with a restricted capability to recognize content. On the one hand, the existing databases are still small, much less than the size of databases in image recognition such as the ImageNet database. In the ImageNet database, there are millions of annotated images, totaling approximately 14 million images with 20 thousand various kinds of objects captured from various angles, perspectives, and environments. However, most of the databases for AT have been built only by the researchers themselves, and the databases are limited in terms of the daily language they can cover, making it difficult to cover other aspects of life. In Yang’s model [13], the dataset size is less than ten thousand, with fewer than ten categories for each classification task. In Jin’s model [12], there are only 1900 elements in their dataset with only two categories in the classification task. Consequently, the existing trained models are less general, and limited to some basic phrases and sentences. On the other hand, the researchers lack the willingness to upload the self-built databases. In consequence, the existing public database for AT is extremely rare, lacking a substantial foundation for training large-scale AT speech recognition models.

In terms of hardware, the flexibility of the current AT circuit is still deficient. In the existing soft wearable instruments, the circuit is always flexible and combined with the sensors. In 2023, Yoo et al., in Rogers’ group, proposed a wireless sensing system for physiological monitoring that integrated the circuit and sensors into a single combination [126]. Similarly, in 2023, Shinjae Kwon et al. also developed a sleep monitoring system that combined the circuits and sensors into a unified assembly [127]. However, current AT often requires pairing with external power supplies, microcontrollers, etc., which hampers their portability and impedes their widespread adoption and practical use. The cooperation relationship between sensors, circuits, and microcontrollers also should be optimized. Furthermore, to enhance stability and achieve a higher signal-to-noise ratio, it is necessary to incorporate facilities and devices designed for shielding against external interference. The ideal solution is a system where the sensors, circuits, microprocessors, etc. are flexible and as small as possible.

In addition to the two aspects mentioned earlier, the existing AT also falls short of meeting the requirements of portable medical products. The current trend in the development of portable medical products is toward multifunctionality, catering to a wide range of operating scenarios. The pursuit of multifunctionality not only enhances efficiency but also reduces costs and resource consumption, expanding the scope of applications. However, the functions of the existing AT are relatively homogeneous and require further improvement to align with the evolving direction of medical device development. In the near future, it is crucial to expand the application scope of the AT. Firstly, we can leverage the AT’s capability to acquire EMG signals for sleep monitoring to diagnose certain diseases. Then, we can enhance our algorithms to enable language translation functionalities, allowing the AT to assist individuals with communication barriers due to language differences, not just limited to these people with larynx diseases.

Additionally, there is another issue that requires attention. There are relatively limited experimental studies on the AT in clinical applications, which leads to a lack of experimental data to support the clinical safety of the AT. In Yang’s study [13], patients wore the AT for a short duration during testing. However, the effects of prolonged wear and environmental factors such as temperature and humidity on the functionality of the AT remain unclear. Additionally, the study only involved one patient as a tester, which may limit the generalizability of the findings. More participants are needed for a more comprehensive and reliable assessment. In the studies conducted by Jin [12], the evaluation of AT was limited to healthy individuals, lacking data from patients and clinical practice settings. Although the intelligent AT technology theoretically has the potential in medical applications, more experimental data is needed to assess its feasibility and safety in practical medical settings.

In conclusion, a comprehensive review of the AT has been demonstrated which consists of the detection, emitting sound, and algorithms for speech recognition. The sensor for detecting sound can be divided into capacitive, piezoelectric, electromagnetic, and piezoresistive. Then some devices for emitting sound, including the TA effect are discussed. The algorithms utilized by AT for speech recognition also has been analyzed carefully. Finally, we state the challenge and outlook of this AT. Compared with the conventional electrolarynx, the AT with flexible material and adhesive to the skin very well is more portable and easier to use for mute people and is superior in other aspects.

## Figures and Tables

**Figure 1 sensors-24-01493-f001:**
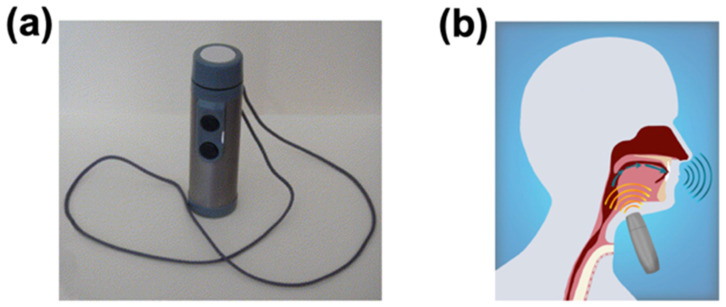
The conventional electrolarynx. (**a**) The overview of the conventional electrolarynx. Reproduced with permission [6]. (**b**) The usage of the conventional electrolarynx [8].

**Figure 2 sensors-24-01493-f002:**
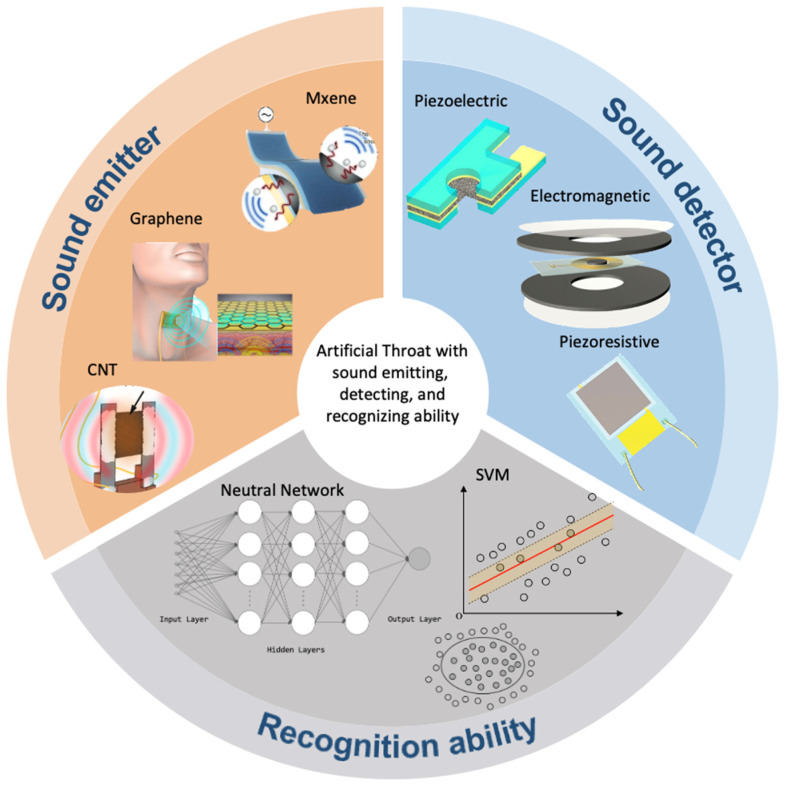
Intelligent flexible AT serves as the sound emitter, detection, and recognition devices [13,16,17,18,19,20].

**Figure 3 sensors-24-01493-f003:**
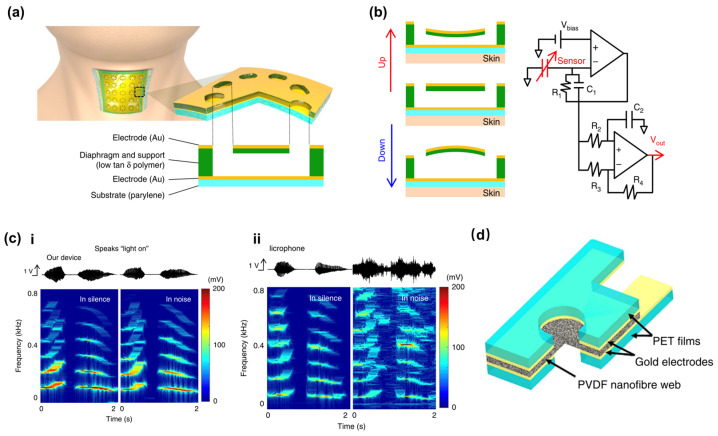
Sound sensor based on capacitive and piezoelectric effect. (**a**) Illustration of the capacitive sound sensor attached to the neck and the diaphragm structure. (**b**) The circuit diagram within the sensor. (**c**) Comparison of waveform and frequency spectrum in silent and noisy environments when a person speaks ‘light on’ with the capacitive sound sensor and licrophone. (**i**) utilizes the capacitive sound sensor, while (**ii**) utilizes the licrophone [33]. (**d**) Sound sensor structure based on the piezoelectric materials [17].

**Figure 4 sensors-24-01493-f004:**
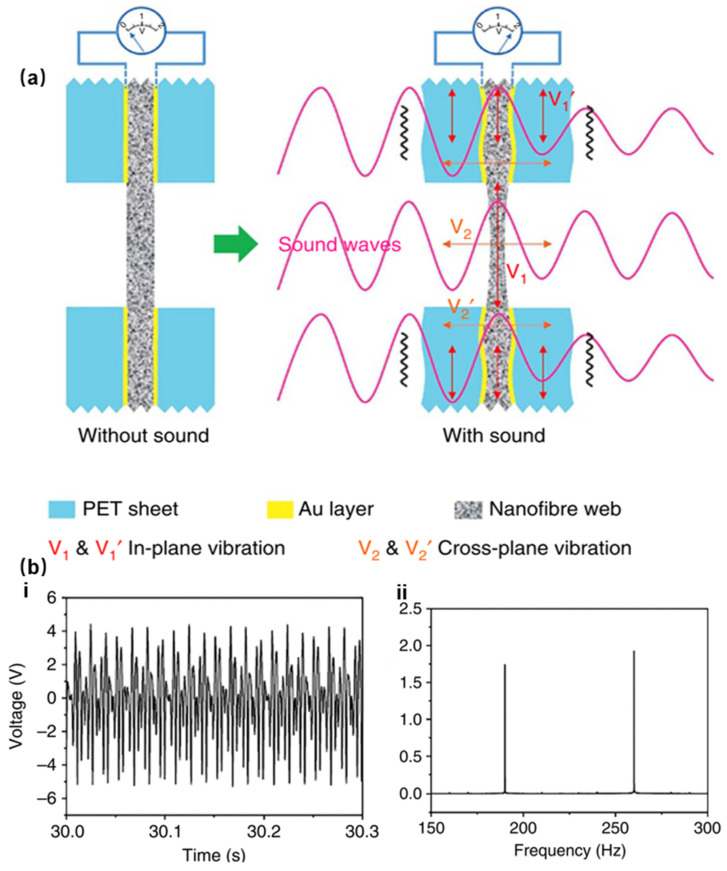
Sound sensor based on piezoelectric effect. (**a**) When sound waves hit the piezoelectric nanofibers, vibration of the piezoelectric materials takes place. (**b-i**) is the voltage spectrum under double-frequency sound waves, while (**b-ii**) is the frequency under double-frequency sound waves [17].

**Figure 5 sensors-24-01493-f005:**
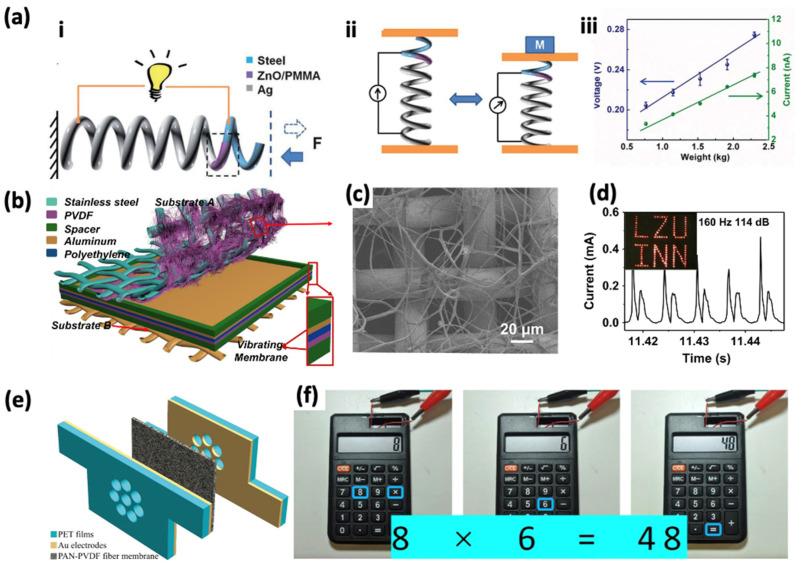
Self-powered piezoelectric sound sensors. (**a-i**) Schematic structure of nanogenerator based on ZnO. (**a-ii**,**a-iii**) The voltage and current vary when weight is put on the nanogenerator sensor [32]. (**b**) Schematic of a fabricated sound TENG. (**c**) SEM image of the PVDF nanofibers. (**d**) 138 LEDs were driven by the sound TENG with the sound of 144 dB and 160 Hz [47]. (**e**) Structure of the PAN-PVDF noise harvester structure. (**f**) The sound sensor powers the calculator to perform the calculation process [48].

**Figure 6 sensors-24-01493-f006:**
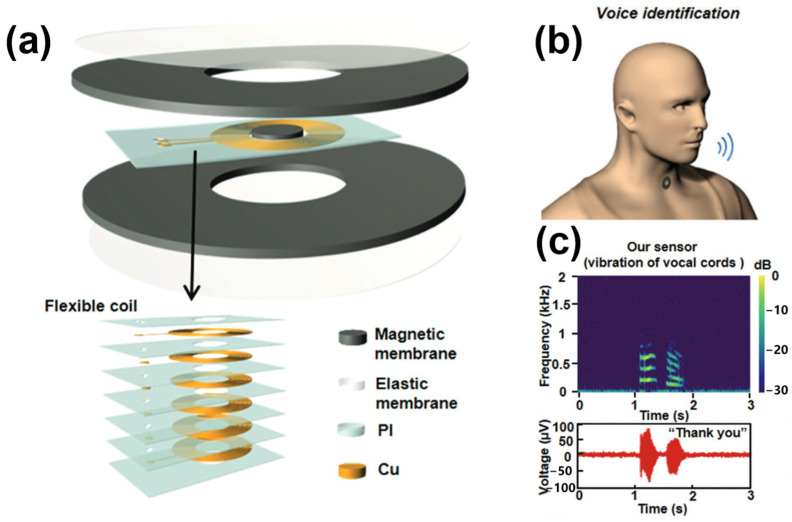
Electromagnetic effect-based sound sensor. (**a**) The structure of the electromagnetic sensor. (**b**) The sensor is attached to the neck for voice identification. (**c**) The time-frequency diagram measured by a sensor attached to the neck and the frequency spectrum converted by a fast Fourier transform [18].

**Figure 7 sensors-24-01493-f007:**
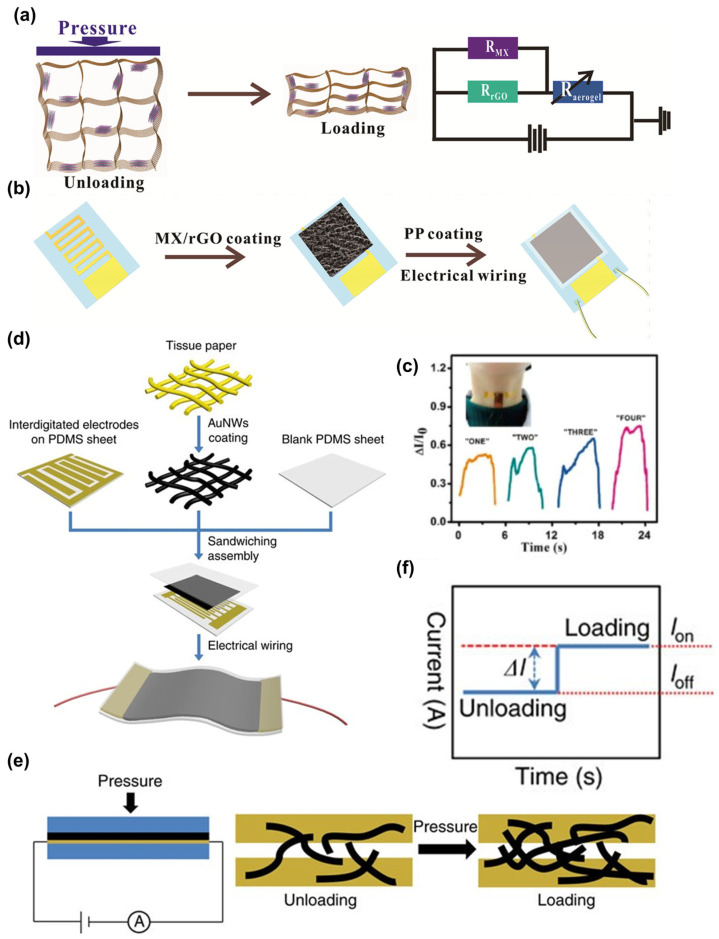
Piezoresistive effect-based sound sensor. (**a**) The schematic illustration of the piezoresistive material sensing mechanism. (**b**) The fabrication process of the MX/rGO sensor. (**c**) The continuous monitoring of the tiny strain and human voice using MX/rGO sensors [19]. (**d**) Schematic illustration of the fabrication of the piezoresistive sensor based on AuNWs. (**e**,**f**) The illustration of the sensing mechanism and current changes when applying pressure [67].

**Figure 8 sensors-24-01493-f008:**
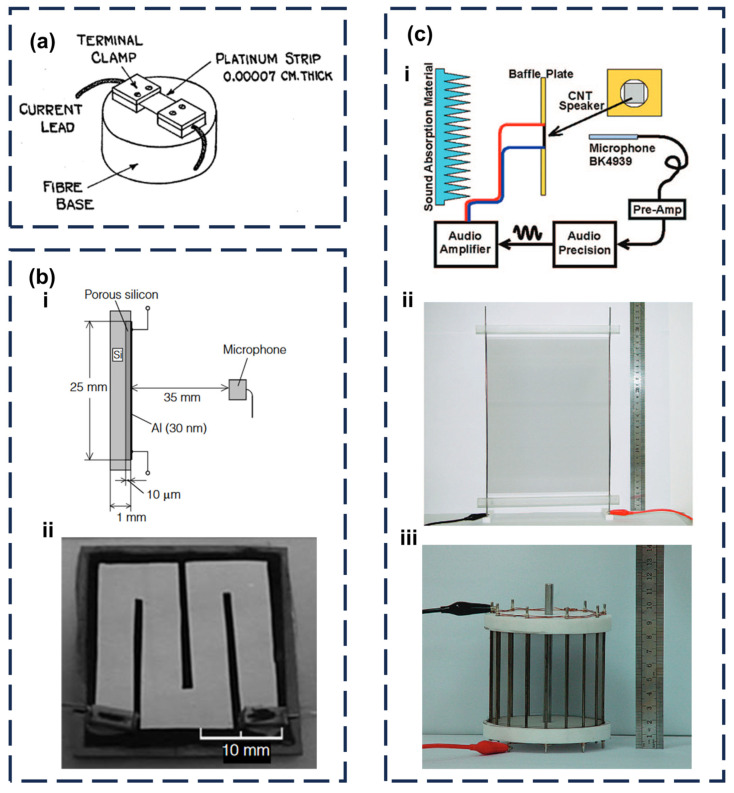
Development of the TA sound emitter. (**a**) Simple TA sound emitter made of the platinum strip [84]. (**b-i**) Cross-sectional view of the fabricated device and set-up for sound measurement. (**b-ii**) Photograph of a top view of the device [81]. (**c-i**) Schematic illustration of the experimental setup for CNT thin film sound emitters. (**c-ii**) A4 paper size CNT thin film sound emitter. (**c-iii**) the cylindrical cage shape CNT thin film sound emitter [83].

**Figure 9 sensors-24-01493-f009:**
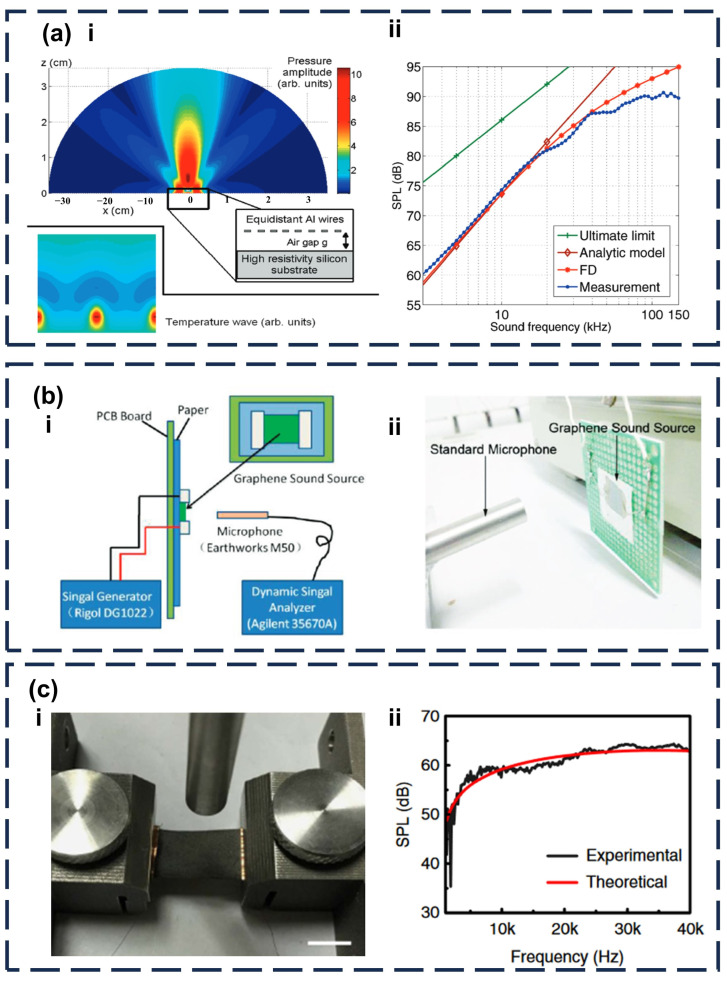
Development of the TA sound emitter. (**a-i**) An illustration of sound radiation from array of metal wires in modeling and experiments of TA sound emitters. (**a-ii**) comparisons between measurement and analytic model [87]. (**b-i**) Schematic diagram of test platform for graphene sound emitter. (**b-ii**) Onsite photo of the experimental setup. [88] (**c-i**) onsite photo of the experimental setup for graphene-based intelligent AT. (**c-ii**) the SPL versus the frequency showing that the model agrees well with experimental results [58].

**Figure 10 sensors-24-01493-f010:**
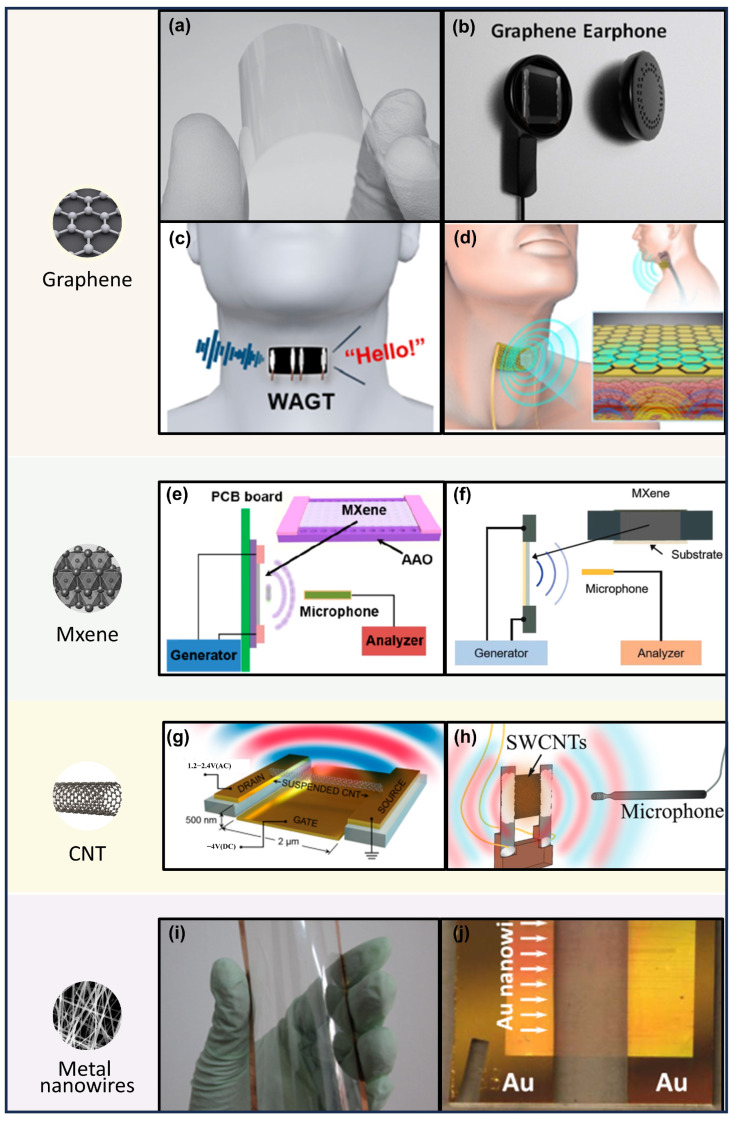
TA sound emitter made of different materials. (**a**) Monolayer graphene on PET as transparent and flexible sound emitters [91]. (**b**) Graphene earphone in a commercial earphone casing [90]. (**c**) Schematic of graphene sound emitter when attached to throat [80]. (**d**) Schematic diagram of the interaction paradigm of the intelligent artificial graphene throat [13]. (**e**) Schematic structure of MXene-based TA sound emitter [93]. (**f**) Schematic of the MXene-based TA sound measurement setup [20]. (**g**) Schematic diagram of suspended CNT-based TA sound emitter geometry [99]. (**h**) Schematic structure of SWCNTs-based TA sound emitter [16]. (**i**) Photograph of flexible and transparent silver nanowire-based sound emitter [100]. (**j**) Optical image of gold nanowire-based TA sound emitter [101].

**Figure 11 sensors-24-01493-f011:**
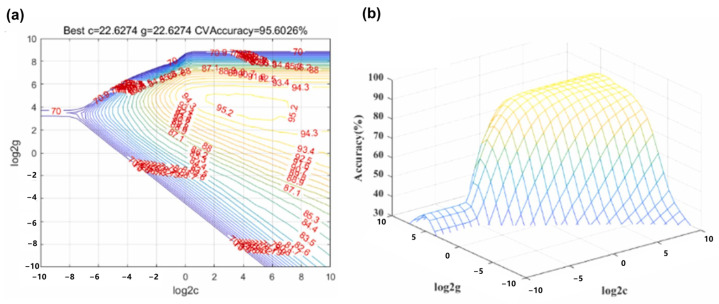
Post-processing and recognition of the detected signals. (**a**,**b**) 3D view and the contour view of SVM parameter selection [114].

**Figure 12 sensors-24-01493-f012:**
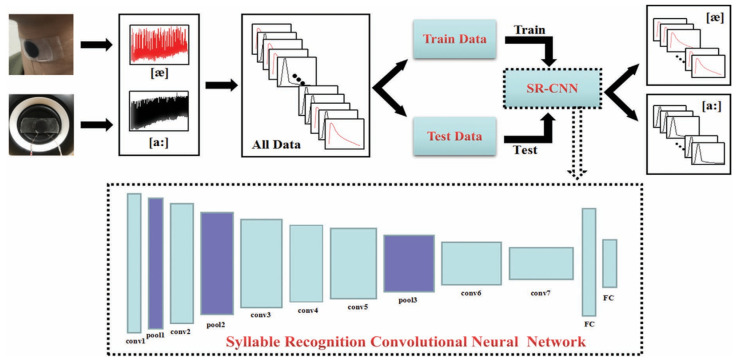
Experimental flow chart and the structure of overall classification by SR-CNN [12]. The SR-CNN is composed of seven convolution layers, three pooling layers, and two fully connected layers.

**Figure 13 sensors-24-01493-f013:**
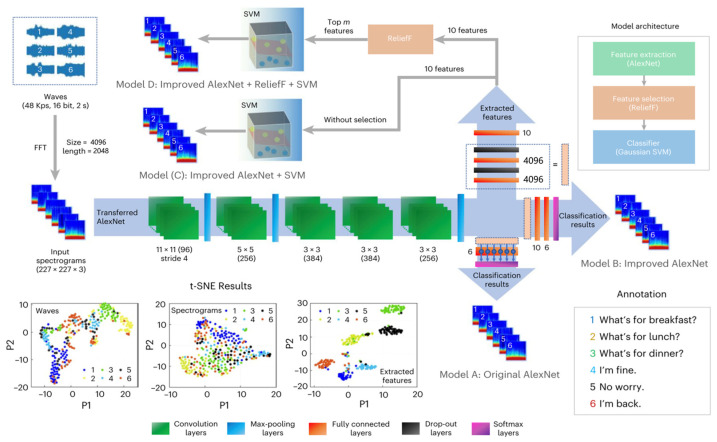
Structure of different integrated models [13]. Model A is the original AlexNet, model B is the improved model, model C is a combination model of two artificial algorithms, improved AlexNet and SVM, and model D is a combination of three artificial algorithms, improved AlexNet, Relief, and SVM.

**Figure 14 sensors-24-01493-f014:**
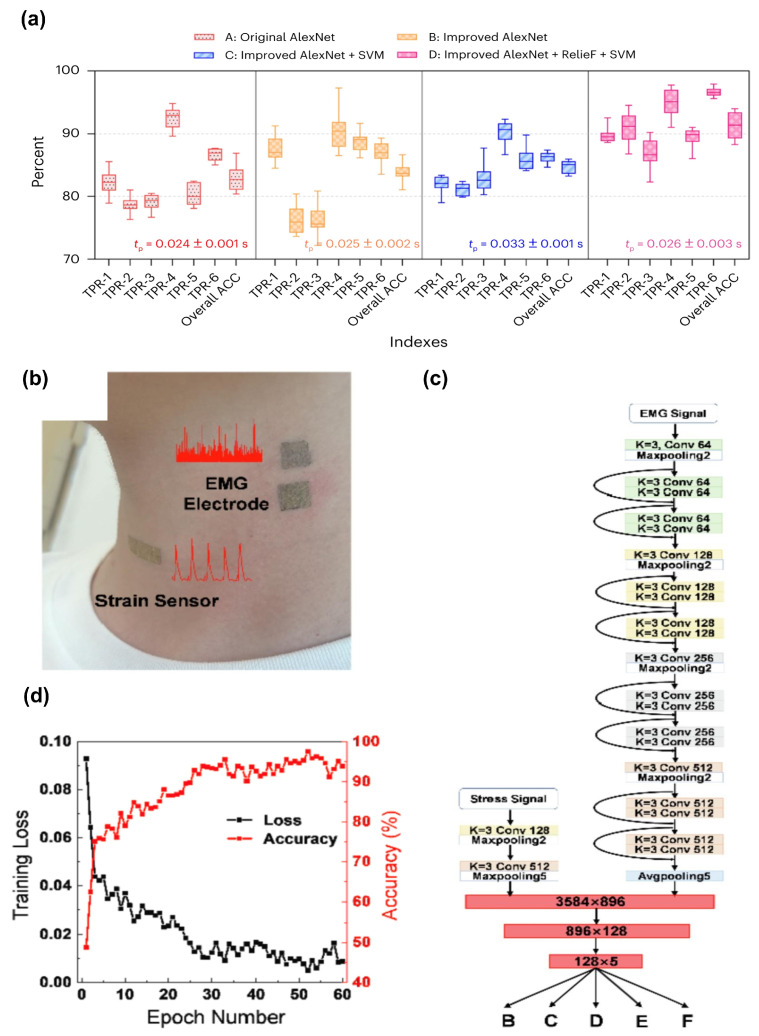
Post-processing and recognition of the detected signals. (**a**) Comparison of the improved AlexNet model with the original AlexNet. ACC, accuracy; tp, time for prediction; TPR, true positive rate [13]. (**b**) The illustration of Au/PU nanomesh strain sensor and Au nanomesh EMG electrodes. (**c**) The SCNN algorithm consists of ResNet18 for the EMG signal and two-layer CNN for the stress signal. (**d**) The training loss and classification accuracy for the SCNN model [15].

**Figure 15 sensors-24-01493-f015:**
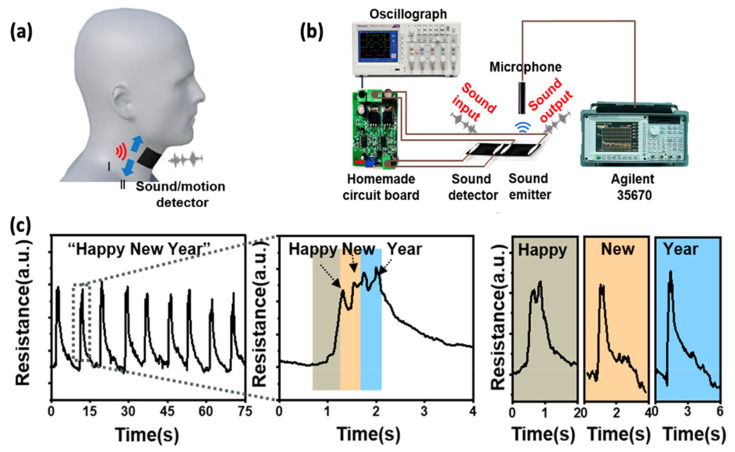
AT is a combination of sound detection, emission, and recognition. (**a**) The AT can serve as a sound and motion sensor. (**b**) The sound detection system. The sound detection device is connected to the circuit board and displays resistance. (**c**) The resistance response to the sound “Happy New Year” [80].

**Figure 16 sensors-24-01493-f016:**
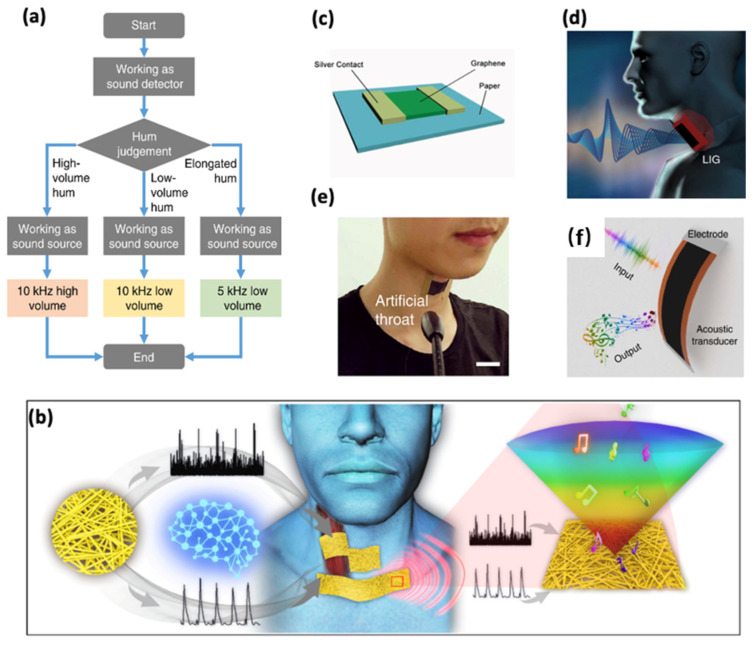
The AT can serve as a sound sensor and emitter with a speech recognition function. (**a**) The working procedure of the artificial throat [58]. (**b**) The composition of the AT based on Au/PVA and Au/PU nanomesh [15]. (**c**) Schematic view of a sound emitter using graphene as the emission component [88]. (**d**) The AT can detect the movement of the throat and emit sound. (**e**) The tester wearing the graphene AT. Scale bar, 1 cm [58]. (**f**) The AT serves as the sound emitter and sound sensor simultaneously [58].

**Table 1 sensors-24-01493-t001:** The device with sound emitting and sound detecting functions.

Material	Substrate	Principle of Sound Emitting	Signal of Sound Detecting	Algorithm	Accuracy	Reference
Graphene	PI	TA effect	Vibration	None	None	[58]
Graphene	PI	TA effect	Vibration	None	None	[80]
Au and Au/PU	PU	TA effect	Vibration and EMG	Synergetic GNN	98.9%	[15]
Graphene	PI	TA effect	Vibration and EMG	CNN	>88.14%	[13]
MXene	Parylene and so on	TA effect	Unable to detect	None	None	[20]
MXene	PDMS	Unable to emit	Vibration	CR-CNN	>70%	[12]
CNT	None	TA effect	Unable to detect	None	None	[16]
CNT	Zn	Unable to emit	Vibration	None	None	[125]

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
