# Peer review of "Intelligent, Flexible Artificial Throats with Sound Emitting, Detecting, and Recognizing Abilities"

_sensors, 2024, doi:10.3390/s24051493_

Round 1
Reviewer 1 Report
Comments and Suggestions for Authors
This review is devoted to an interesting and very practically important problem of creating an artificial throat. The authors of the review did a great job, considering three important aspects of creating such a device: sound emitting, sound detecting and sound recognizing. All these aspects are considered with sufficient details. Thus, this work provides good insight into existing approaches to creating an artificial throat. However, I would like to note that there is another very interesting approach for sound recognition, namely “Silent speech interfaces” [https://doi.org/10.1016/j.specom.2009.08.002]. In the presented review, this approach is referred to in section 2.5 as sound detector, although it should refer to speech recognition. A small drawback of the work is that the illustrations are too overloaded with a lot of small text. This is especially true for Figure 3, Figure 7, Figure 10 and Figure 11. Perhaps they should be divided into parts and the scale of each illustration should be increased.
In general, this review can be recommended for publication after these minor issues have been corrected.
Reviewer 2 Report
Comments and Suggestions for Authors
The authors provide a comprehensive review of intelligent flexible artificial throats (AT) with sound emitting, detecting, and recognizing abilities. There are several challenges that need to be addressed in the development of AT. Firstly, the current artificial algorithms used for AT are limited in their generalizability and dataset size. Additionally, the existing databases for AT are small and limited in their coverage of daily language, making it difficult to train large-scale speech recognition models. Furthermore, the flexibility of the current AT circuit is deficient, requiring external power supplies and hindering portability. Moreover, there is a lack of experimental data to support the clinical safety of AT in medical applications.
